# Changes in soft drinks purchased by British households associated with the UK soft drinks industry levy: a controlled interrupted time series analysis

Nina T Rogers,[1] David Pell [ID],[1] Oliver T Mytton [ID],[1,2] Tarra L Penney,[3] Adam Briggs,[4,5] Steven Cummins [ID],[6] Catrin Jones,[1] Mike Rayner,[4,7] Harry Rutter [ID],[8] Peter Scarborough [ID],[7,9] Stephen Sharp,[1] Richard Smith,[10] Martin White,[1] Jean Adams [ID][1]

For numbered affiliations see end of article.

**Correspondence to**
Dr Jean Adams;
jma79@medschl.cam.ac.uk

## ABSTRACT

**Objective** To determine changes in household purchases of drinks 1 year after implementation of the UK soft drinks industry levy (SDIL).

**Design** Controlled interrupted time series.

**Participants** Households reporting their purchasing to a market research company (average weekly n=22 091), March 2014 to March 2019.

**Intervention** A two-tiered tax levied on soft drinks manufacturers, announced in March 2016 and implemented in April 2018. Drinks with ≥8 g sugar/100 mL (high tier) are taxed at £0.24/L, drinks with ≥5 to <8 g sugar/100 mL (low tier) are taxed at £0.18/L.

**Main outcome measures** Absolute and relative differences in the volume of, and amount of sugar in, soft drinks categories, all soft drinks combined, alcohol and confectionery purchased per household per week 1 year after implementation.

**Results** In March 2019, compared with the counterfactual, purchased volume of high tier drinks decreased by 140.8 mL (95% CI 104.3 to 177.3 mL) per household per week, equivalent to 37.8% (28.0% to 47.6%), and sugar purchased in these drinks decreased by 16.2 g (13.5 to 18.8 g), or 42.6% (35.6% to 49.6%). Purchases of low tier drinks decreased by 170.5 mL (154.5 to 186.5 mL) or 85.8% (77.8% to 93.9%), with an 11.5 g (9.1 to 13.9 g) reduction in sugar in these drinks, equivalent to 87.8% (69.2% to 106.4%). When all soft drinks were combined irrespective of levy tier or eligibility, the volume of drinks purchased increased by 188.8 mL (30.7 to 346.9 mL) per household per week, or 2.6% (0.4% to 4.7%), but sugar decreased by 8.0 g (2.4 to 13.6 g), or 2.7% (0.8% to 4.5%). Purchases of confectionery and alcoholic drinks did not increase.

**Conclusions** Compared with trends before the SDIL was announced, 1 year after implementation, volume of all soft drinks purchased combined increased by 189 mL, or 2.6% per household per week. The amount of sugar in those drinks was 8 g, or 2.7%, lower per household per week. Further studies should determine whether and how apparently small effect sizes translate into health outcomes.

---

### STRENGTHS AND LIMITATIONS OF THIS STUDY

⇒ We used a large, nationally representative dataset, included a control category, and explored changes in two potential substitute categories (alcohol and confectionery).

⇒ We only included purchases brought into homes.

⇒ We did not assess changes in other categories beyond soft drinks, alcohol and confectionery.

⇒ The estimate of effect size in interrupted time series analyses is based on a modelled counterfactual that might be inaccurate.

⇒ Attribution of effects in interrupted time series analyses is vulnerable to time varying confounding such as cointerventions.

---

**Trial registration number** ISRCTN18042742.

## INTRODUCTION

High consumption of sugar sweetened beverages (SSBs) is associated with increased risk of dental caries, obesity, type 2 diabetes and cardiovascular disease.[1–3] The WHO recommends the use of SSB taxes to reduce consumption.[4] A systematic review of studies published in June 2018 suggests that SSB taxes lead to decreases in the sales, purchasing and consumption of taxed drinks.[5] More recent findings support this conclusion.[6–10] Although price is one important mediator of these changes,[11–16] other potential mechanisms include reformulation of products to reduce sugar concentration, smaller portion sizes, and increases in the perception of SSBs being harmful to health associated with them being grouped with other taxed products such as alcohol and tobacco.[17] Furthermore, any public health benefits of reduced SSB

BMJ

---

## Box 1    Glossary of terms

*Soft drinks industry levy (SDIL)*—a tiered tax on manufacturers of sugar sweetened beverages.

*Levy exempt drinks*—drinks exempt from the SDIL irrespective of sugar content; that is, drinks containing >75% milk, drinks containing >1.2% alcohol, and drinks sold as alcohol replacements, drinks sold as powders, 100% fruit juices, and drinks sold by manufacturers selling less than 1 million litres of drinks not exempt for other reasons each year.

*High tier drinks*—drinks that are not levy exempt and contain ≥8 g of sugar per 100 mL.

*Low tier drinks*—drinks that are not levy exempt and contain ≥5 g to <8 g of sugar per 100 mL.

*No levy drinks*—drinks that are not levy exempt but contain <5 g of sugar per 100 mL; we subdivided this category into drinks containing >0 to <5 g of sugar per 100 mL, drinks containing 0 g of sugar per 100 mL. Bottled water was considered separately.

*Levy liable drinks*—drinks that are not levy exempt drinks; that is, the sum of high tier drinks, low tier drinks and no levy drinks.

*Soft drinks*—any drink not containing alcohol.

*Confectionery*—products in the sugar confectionery and chocolate confectionery categories.

*Toiletries*—products in the shampoo, hair conditioner and liquid soap categories.

---

consumption associated with SSB taxes might be negated by increased consumption of substitutes such as confectionery and alcohol.[18–20]

The UK soft drinks industry levy (SDIL) was one of the first taxes on SSBs explicitly designed to incentivise manufacturers of SSBs to reduce sugar content.[21 22] This is reflected in three design features. First, the SDIL is levied on manufacturers, importers and bottlers rather than on consumers. Second, the levy includes two tiers: £0.24/L for drinks containing ≥8 g total sugar per 100 mL, and £0.18/L for drinks containing ≥5 g and <8 g total sugar per 100 mL. Third, the SDIL was intentionally announced in 2016, 2 years before implementation in 2018, to allow manufacturers time to adjust. The SDIL also provides exemptions (box 1).[23]

Two before and after analyses have shown reductions of around 30% in sales weighted sugar concentration of levy eligible drinks in the UK from before the announcement of the SDIL on 16 March 2016 to after implementation on 6 April 2018.[24 25] However, background trends in purchases of sugary drinks are not stable, with decreases reported over several years.[26] This makes it difficult to attribute before and after decreases in sugary drinks purchases to the SDIL. An interrupted time series analysis found that the announcement and implementation of the SDIL were together associated with a 34 percentage point reduction in the proportion of levy liable drinks with >5 g total sugar per 100 mL, indicating substantial reformulation of the market.[15] Changes in prices across the UK soft drink market were also reported, although it was difficult to discern clear patterns in these, with some levied categories increasing and others decreasing in price. In a controlled interrupted time series analysis including data

up to the point of SDIL implementation, we found that the SDIL announcement was associated with changes in both the volume of, and sugar purchased in, drinks in many categories.[27] Overall we found no change in total volume of purchases of all soft drinks combined, but a small increase in sugar purchased from soft drinks of 5.3 g per household per week, or 1.7%.

In this paper, our aim was to determine whether household purchases of drinks and confectionery had changed 1 year after implementation of the SDIL.

## METHODS

Here, we extend our previous analyses[27] to study changes in the volume of, and amount of sugar in, household purchases of drinks in each levy tier, exempt drinks categories (including alcoholic drinks), and confectionery from 2 years before the announcement of the SDIL to 1 year after its implementation (March 2014 to March 2019). As before, we used controlled interrupted time series methods, with toiletries included as a control category.[27] We compared observed changes associated with the announcement and implementation of the SDIL to the counterfactual scenarios in which the announcement and implementation did not take place. Including a full 2 years of data before the announcement enables us to estimate preintervention trends and project these forward as counterfactual scenarios. The protocol is published elsewhere[28] and the study was registered. This study is reported in accordance with the Strengthening the Reporting of Observational Studies in Epidemiology guideline (see online supplemental material A).

### Data source

We used data from a panel of households reporting their purchasing on a weekly basis to a market research company (Kantar Worldpanel; KWP). Participating households are asked to record all food and drink purchases brought into the home (including those ordered online and delivered) through barcodes scanners and manual report. Purchasing information is uploaded weekly, where it is linked to nutritional data collected by KWP field workers on a rolling basis. Households record their personal characteristics and receive gift vouchers worth about £100 ($122; €112) annually—equivalent to 0.3% of median UK annual household income after tax in 2019 (£29 600).[29]

KWP samples households from across Great Britain (GB) using proprietary methods, aiming to achieve a sample that is demographically representative of GB households. Data exclude households that record fewer than six purchases weekly along with those whose adjusted weekly spend is lower than an undisclosed minimum. KWP applies proprietary weights to purchases to adjust for these exclusions and maintain the representativeness of the panel. We used these weights throughout.

The main data cleaning that occurred before analysis involved assigning products and product groups in the

KWP dataset to SDIL relevant groups. This was done based on KWP assigned product groups, product names and nutritional content. In previous work we found some evidence of error, but not bias, in the sugar concentration reported by KWP compared with information provided on manufacturers' websites.[27]

### Product categories: drinks, confectionery and toiletries

Purchased drinks that were levy liable were divided into high tier, low tier or no levy based on sugar content (see box 1 for definitions). No levy drinks were additionally disaggregated, as described in box 1.

As the SDIL might have led to substitution to other drinks categories, we also examined purchasing of levy exempt drinks in several categories: milk based drinks (comprising milk, milk alternatives such as soya drinks, and yoghurt based juices and drinks), alcoholic drinks (comprising both alcoholic and alcohol replacement drinks), no added sugar fruit juices, and drinks sold as powder (eg, tea, coffee, hot chocolate). Other exempt categories (infant formulas and drinks sold for medical purposes) were excluded.

We also hypothesised that the SDIL might lead to substitution from sugary drinks to other high sugar categories. To investigate this, we used sugar and chocolate confectionery purchases (referred to as confectionery).

### Control group

To control for background trends in household purchases we used purchases of shampoo, hair conditioner and liquid soap (ie, toiletries). Toiletries meet the proposed criteria for a controlled interrupted time series: they are robust to seasonality and may have similar purchase volumes by households regardless of socioeconomic position or other potential confounders.[30]

### Outcome measures

Most evaluations of SSB taxes focus on volume of drinks purchased. However, the SDIL's focus on reformulation makes the sugar purchased in drinks of additional public health interest. Thus, the outcome measures of interest were mean volume purchased per household per week in each of the drink categories and grams per household per week of confectionery; and mean sugar purchased per household per week from each of the drink categories

and confectionery. Data were aggregated at the weekly level and analysed as a time series.

### Overall analysis strategy

Previous evidence indicates that reformulation occurred after the announcement of the SDIL and price changes after implementation.[15] As such, we hypothesised the SDIL might act as two linked interventions: the announcement on 16 March 2016 and implementation on 6 April 2018.[17] Thus, our analysis strategy involved three separate comparisons that isolate the announcement and implementation of the SDIL and then examine the combined effect (figure 1). In the first analysis, we isolated the announcement of the SDIL. Here, we compared anticipatory effects on purchasing 2 years after the announcement to the counterfactual estimated from purchasing in the 2 years before the announcement. This replicates and updates our previous analysis[27] as we anticipate that the stabilising effect of including additional postannouncement data likely reduces error. In the second analysis, we isolated the implementation of the SDIL. Here, we compared purchasing 1 year after implementation to the counterfactual estimated from purchasing in the 4 years before implementation. In the third analysis, we considered both the announcement and the implementation and we compared purchasing 1 year after implementation (ie, 3 years after announcement) to the counterfactual estimated from purchasing in the 2 years before the announcement.

Throughout, we used the proprietary weights provided by KWP.

### Primary analysis: category specific analyses

For each of the three analyses we developed separate controlled interrupted time series models for volume and sugar purchased from each levy liable and levy exempt drinks category and confectionery (figure 1). Online supplemental material B provides the full model specification.

We present absolute and relative differences between observed purchasing and counterfactual scenarios in the final week of each observation period, with SEs used to calculate 95% CIs for the relative difference obtained using the delta method.[31]

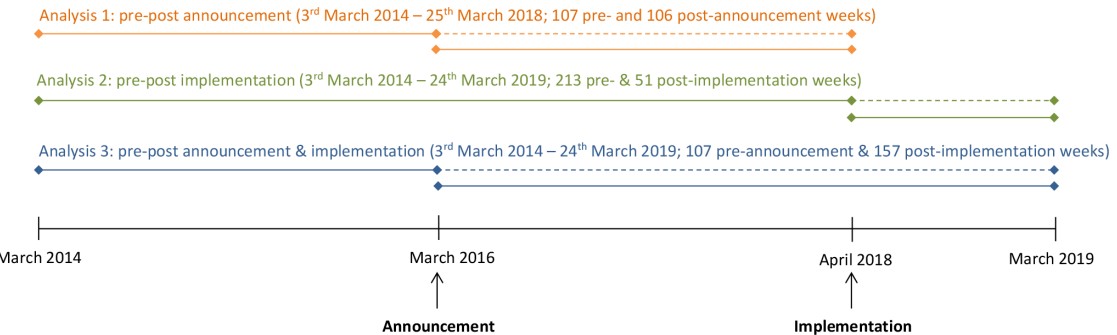

**Figure 1** Schematic of overall analysis strategy. Solid lines indicate observed data; dashed lines indicate counterfactual estimated from previous observed data.

### Secondary analysis: all soft drinks categories combined, irrespective of levy eligibility

Levy exempt drinks include drinks that might contain comparable amounts of sugar to levy liable products. To examine the extent to which the SDIL impacted on the purchased volume of, and amount of sugar in, soft drinks, regardless of SDIL liability, we carried out controlled interrupted time series analysis, combining purchases of all soft drinks irrespective of sugar content (ie, high tier, low tier, no levy, milk and milk based drinks, no added sugar fruit juice and drinks sold as powders), levy liable drinks irrespective of sugar content (ie, high tier, low tier and no levy drinks) and according to sugar content based on levy tiers irrespective of levy eligibility (ie, all soft drinks with ≥8 g of sugar per 100 mL, all soft drinks with ≥5 g to <8 g of sugar per 100 mL and all soft drinks with <5 g of sugar per 100 mL).

### Sensitivity analysis: excluding small manufacturers

The SDIL exempts drinks from manufacturers and producers who sell less than 1 million litres of levy liable drinks annually. As we were unable to obtain a list of exempt manufacturers, our main analyses include all manufacturers. We conducted sensitivity analyses to examine the effect of excluding manufacturers who we estimated to be small. The total purchase volume was summed by manufacturer by year across the 5 years in the KWP dataset, and a mean purchase volume per year for each manufacturer was calculated. In the first sensitivity analysis, we excluded manufacturers with a mean of less than 1 million litres purchased per year. Acknowledging KWP data excludes purchases not brought home, we repeated these analyses excluding manufacturers with mean annual purchased volumes of <0.5 million litres in KWP. We were unable to access accurate estimates of the proportion of all drinks purchases brought home. This value reflects an arbitrary, but we think conservative, estimate of the minimum proportion of drinks brought home.

### Sensitivity analysis: interrupted time series without a control category

Toiletries were chosen as a control condition a priori to account for background trends in household purchases. It is, however, possible that a more appropriate control exists. As we only have access to data on purchasing of the categories described here (confectionery, drinks, toiletries), we were not able to examine alternative potential control categories. To examine the effect of the decision to use toiletries as the control category, we performed an additional sensitivity analysis with no control condition.

### Changes to the protocol

We made several changes to the published protocol.[28] KWP provided additional data that allowed us to increase the precision of our estimates. Specifically, we were able to increase the preannouncement study period from 104 to 107 weeks and reduce the unit of analysis from purchases every 4 weeks to purchases every week. We originally intended to include purchases not brought home. We excluded these purchases, however, as these data were not available before mid-2015, meaning that robust preannouncement trends could not be estimated. Although we originally intended to combine all no levy drinks, we present these disaggregated into those with >0 g and <5 g of sugar per 100 mL and 0 g of sugar per 100 mL, as well as bottled water, as trends for these different categories are noticeably different. Our original intention to explore potential disparities across socioeconomic groups will be pursued in future work.

### Patient and public involvement

The steering group for the wider SDIL evaluation includes two lay members and meets two times a year. Patients and the public were not involved in developing the research question, the outcome measures, the design or the conduct of the work reported here. The steering group has regularly contributed ideas for routes to dissemination.

### Correction of Pell *et al* (2021)

This paper is a corrected version of Pell *et al*,[32] now retracted, which was originally published in the *BMJ*. The analysis presented in the original Pell *et al*[32] paper included an incorrect weighting variable. This variable was incorrectly calculated as the inverse of what it should have been. The variable was also redundant to the analysis as it replicatsed a component of a second weighting variable also included (the 'proprietary weights provided by KWP' mentioned earlier). The current corrected version replicates the original analysis without this redundant and incorrectly calculated weighting variable. The second, correct, weighting variable (the 'proprietary weights provided by KWP" mentioned earlier) remains included. The authors identified the error themselves and alerted the journal and readers.[33]

### RESULTS

About 31 million purchases of drinks, confectionery and toiletries from March 2014 to March 2019 were included from a mean of 22 091 households each week. The characteristics of included households remained consistent over the study period, and after weighting they largely reflected households in 2014–2019 in the UK (see table 1 in online supplemental material C).

Table 1 summarises households' weekly purchased volumes of, and amounts of sugar in, drinks and other categories over the study period. Substantial reductions in volume of, and sugar in, purchases of SDIL liable drinks were observed in the high and low tiers over time. These reductions were accompanied by a smaller increase in volume of no levy drinks purchased, but proportionally much greater increases in sugar purchased in these drinks.

**Table 1** Mean volume of, and amount of sugar in, purchased drinks and confectionery per household per week in relation to the UK soft drinks industry levy, March 2014 to March 2019 (weighted)

| | Mean (SD) volume (mL) per household/week | | | Mean (SD) amount of sugar (g) per household/week | | |
|---|---|---|---|---|---|---|
| | Preannouncement: March 2014–March 2016 | Postannouncement: March 2016–March 2018 | Post implementation: April 2018–March 2019 | Preannouncement: March 2014–March 2016 | Post announcement: March 2016–March 2018 | Post implementation: April 2018–March 2019 |
| **Levy liable drinks (sugar/100 mL)** | | | | | | |
| High tier (≥8 g) | 880 (128) | 680 (136) | 363 (76) | 98 (14) | 76 (15) | 40 (9) |
| Low tier (≥5 g to <8 g) | 155 (32) | 147 (37) | 75 (32) | 10 (2) | 10 (2) | 5 (2) |
| No levy (<5 g) | 1811 (169) | 1876 (216) | 2448 (321) | 12 (2) | 12 (3) | 25 (5) |
| >0 g to<5 g | 785 (78) | 768 (92) | 989 (139) | 12(2) | 12(3) | 25(5) |
| 0 g | 1027 (104) | 1108 (132) | 1459 (190) | 0 (0) | 0 (0) | 0 (0) |
| Bottled water | 591 (72) | 714 (90) | 786 (138) | 0 (0) | 0 (0) | 0 (0) |
| **Levy exempt drinks** | | | | | | |
| Alcoholic drinks* | 1874 (380) | 1872 (456) | 1948 (467) | . | . | . |
| Milk and milk based drinks† | 3546 (137) | 3540 (155) | 3542 (148) | 172 (7) | 172 (8) | 170 (7) |
| Fruit juices with no added sugar | 516 (29) | 502 (44) | 520 (47) | 51 (3) | 49 (4) | 50 (5) |
| Drinks sold as powders (g) | 95 (12) | 88 (11) | 90 (11) | 21 (3) | 19 (3) | 18 (3) |
| **Other categories** | | | | | | |
| Confectionery (g) | 308 (91) | 303 (92) | 318 (100) | 173 (51) | 170 (52) | 178 (57) |
| Toiletries | 123 (8) | 120 (8) | 121 (9) | . | . | . |
| All soft drinks combined (ie, excluding alcohol) | 7595 (295) | 7547 (466) | 7826 (540) | 364 (17) | 337 (24) | 307 (19) |

*Sugar from alcoholic drinks is not included here as many alcoholic drinks contain sugar but the product label does not provide the amount.
†Milk comprises drinks in the following categories: semi-skimmed; specific low fat % milk (eg, 1% fat milk); whole milk; buttermilk; modified milk; other milk; other non-cows milk; rice drink; soya milk. Skimmed milk is excluded from all analysis in this paper due to missing data.

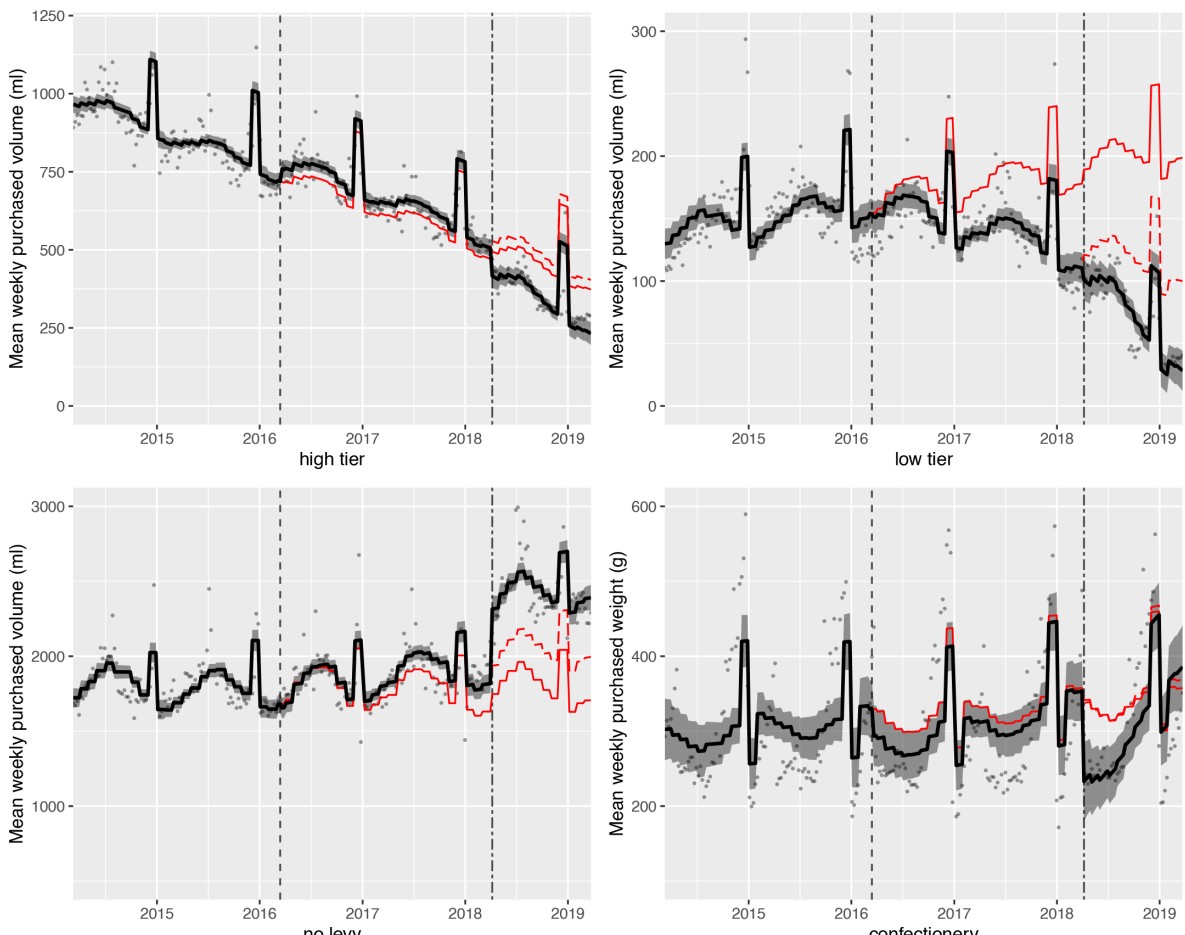

**Figure 2** Observed and modelled volume (mL) of drinks liable to the soft drinks industry levy (SDIL), and weight of confectionery (g) purchased per household per week, March 2014 to March 2019 (weighted). Points are observed data for drinks/confectionery; black lines (with shadows) show modelled data (and 95% CIs); red lines indicate the counterfactuals had the announcement (red solid line) and implementation (red dashed line) not happened; the first dashed vertical line indicates the announcement of SDIL; the second dashed vertical line indicates the implementation of SDIL; the Y axis varies in scale between panels to maximise the resolution of figures; modelled purchases include averaged effects for seasonality and the impact of December and January (Christmas period), and for confectionery, Easter; the control category of toiletries is shown in figure 3.

## Primary analysis: category specific results

Results of the controlled interrupted time series analyses of purchased volume of, and sugar in, levy liable drinks and confectionery are shown in figure 2 (volume) and figure 3 (sugar). Absolute and relative changes are summarised in tables 2 and 3. Tables 2a,b in online supplemental material D show level and trend changes from these models. Figures 1a,b in online supplemental material D show similar figures and data for subcategories of no levy drinks, bottled water and exempt categories.

### High tier drinks

The trend in purchased volume of, and sugar in, high tier drinks continued downwards throughout the study period. The announcement of the SDIL was associated with an increase in purchased volume of (34.7 mL (95% CIs 8.1 to 61.4 mL, or 7.3% (1.7% to 12.9%)), and sugar in (5.5 g (3.8 to 7.2), or 10.8% (7.4% to 14.1%)), these drinks. In contrast, the implementation of the SDIL was associated with a reduction in purchased volume of, and sugar in, these drinks. The volume of high tier drinks

purchased was 171.6 mL (135.1 to 208.1 mL) per household per week, or 42.5% (33.5% to 51.6%), lower in March 2019 compared with the counterfactual estimated from pre-implementation trends. The reductions associated with implementation outweighed the increases associated with announcement, such that the intervention as a whole was associated with a decrease in purchased volume of 140.8 mL (104.3 to 177.3 mL) per household per week or 37.8% (28.0% to 47.6%) and sugar of 16.2 g (13.5 to 18.8 g) per household per week or 42.6% (35.6% to 49.6%) from these drinks.

### Low tier drinks

Purchased volume of, and sugar in, low tier drinks gradually increased before the announcement of SDIL. The announcement was associated with a reversal of this trend. There were reductions in purchased volume of, and sugar in, low tier drinks associated with announcement, implementation and the whole intervention. Compared with the counterfactual estimated from preannouncement trends, in March 2019 the volume of purchased low tier

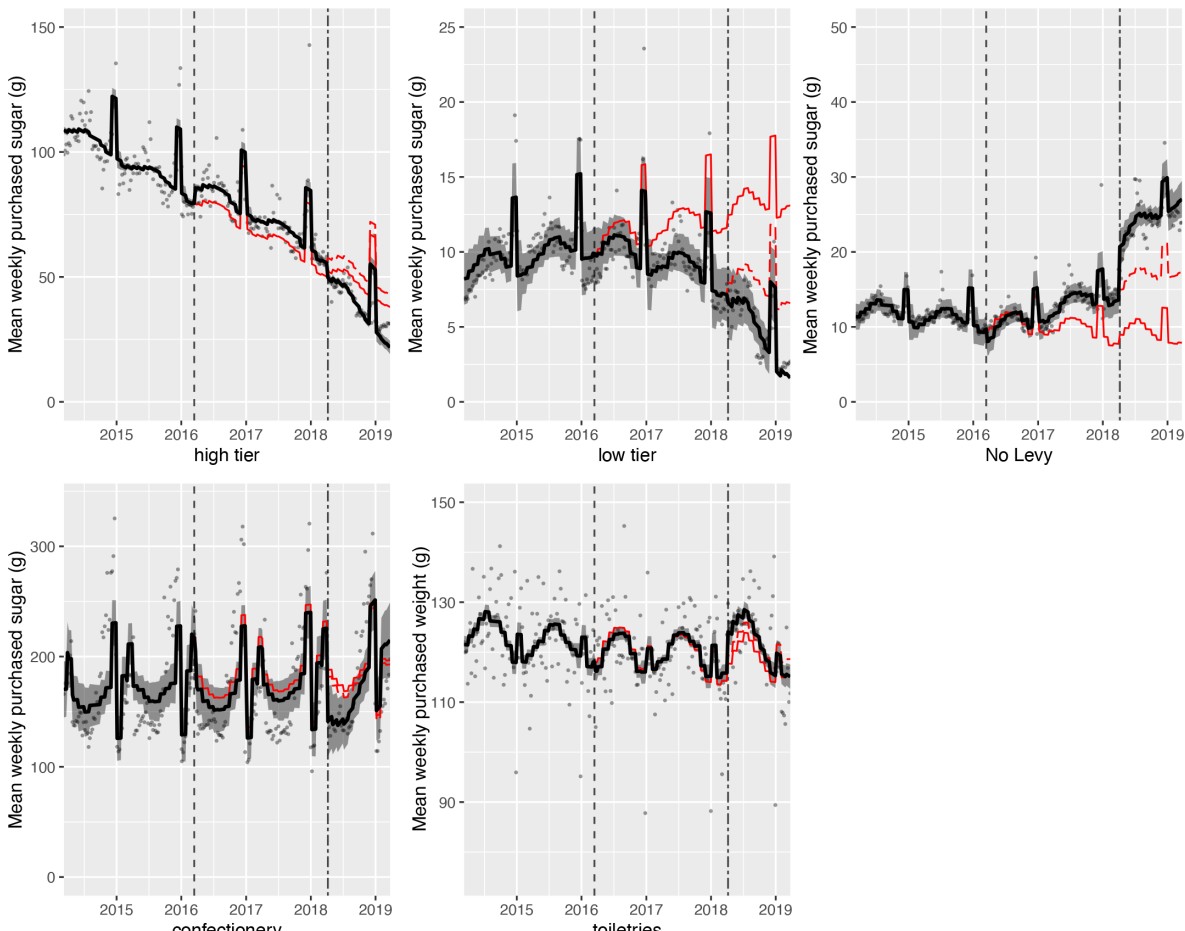

**Figure 3** Observed and modelled amount of sugar (g) in drinks liable to the soft drinks industry levy (SDIL) and confectionery purchased per household per week, March 2014 to March 2019 (weighted). Points are observed data for drinks/ confectionery and toiletries; black lines (with shadows) show modelled data (and 95% CIs); red lines indicate the counterfactuals had the announcement (red solid line) and implementation (red dashed line) not happened; the first dashed vertical line indicates the announcement of SDIL; the second dashed vertical line indicates the implementation of SDIL; the Y axis varies in scale between panels to maximise the resolution of figures; modelled purchases include averaged effects for seasonality and the impact of December and January (Christmas period), and for confectionery, Easter.

drinks per household per week decreased by 170.5 mL (154.5 to 186.5 mL), or 85.8% (77.8% to 93.9%); and sugar purchased in these drinks decreased by 11.5 g (9.1 to 13.9 g) per household per week, or 87.8% (69.2% to 106.4%).

### No levy drinks

Before the announcement of the SDIL there was a gradual upward trend in volume of purchased no levy drinks but a gradual downward trend in purchased sugar. Announcement, implementation and the whole intervention were associated with increases in volume of no levy drinks purchased as well as sugar purchased from those drinks. Overall, purchased volume of no levy drinks in March 2019 was 685.5 mL (599.8 to 771.1 mL) higher, equivalent to 40.2% (35.2% to 45.2%) increase compared with the counterfactual of preannouncement trends. Equivalent figures for sugar purchased from no levy drinks were a 19.2 g (16.7 to 21.6 g) per household per week, equivalent to 242.8% (211.9% to 273.7%), increase.

The implementation and the announcement of the SDIL were associated with increases in volume of purchased drinks with no sugar and with >0 to <5 g total sugar per 100 mL, and increases of sugar in drinks with >0 to <5 g sugar per 100 mL.

### Bottled water

The implementation, but not the announcement, of the SDIL were associated with significant decreases in bottled water purchased which led to an overall decrease in volume of bottled water purchased as a result of the whole intervention of 130.5 mL (88.8 to 174.1 mL) per household per week, or 15.7% (10.4% to 20.9%).

### Levy exempt drinks and confectionery

Overall, the combined announcement and implementation of the SDIL were associated with decreases in purchased volume of alcoholic and milk and milk-based drinks, but no change in sugar purchased from levy exempt categories or from confectionery. Compared

**Table 2** Absolute and relative change in volume of drinks (mL) and confectionery (g) purchased per household per week in relation to the UK soft drinks industry levy, March 2014 to March 2019 (weighted)

| | Analysis 1: preannouncement and postannouncement (March 2014–March 2018) | | Analysis 2: pre-implementation (March 2016–March 2019) implementation | | Analysis 3: preannouncement, postannouncement and pre-implementation, post implementation (March 2014–March 2019) | |
| --- | --- | --- | --- | --- | --- | --- |
| | Absolute change (mL or g) | Relative change (%) | Absolute change (mL or g) | Relative change (%) | Absolute change (mL or g) | Relative change (%) |
| Levy liable drinks (sugar/100 mL) | | | | | | |
| High tier (≥8 g) | **34.7 (8.1 to 61.4)** | **7.3 (1.7 to 12.9)** | **−171.6 (−208.1 to −135.1)** | **−42.5 (−51.6 to −33.5)** | **−140.8 (−177.3 to −104.3)** | **−37.8 (−47.6 to −28.0)** |
| Low tier (≥5 g to <8 g) | **−65.7 (−77.5 to −53.8)** | **−37.1 (−43.7 to −30.4)** | **−71.8 (−87.8 to −55.8)** | **−71.8 (−87.8 to −55.8)** | **−170.5 (−186.5 to −154.5)** | **−85.8 (−93.9 to −77.8)** |
| No levy (<5 g) | **181.0 (118.4 to 243.5)** | **11.1 (7.3 to 14.9)** | **395.0 (309.4 to 480.7)** | **19.8 (15.5 to 24.1)** | **685.5 (599.8 to 771.1)** | **40.2 (35.2 to 45.2)** |
| >0 g to<5 g | **103.8 (75.2 to 132.5)** | **16.7 (12.1 to 21.3)** | **202.0 (162.7 to 241.2)** | **25.0 (20.1 to 29.9)** | **374.6 (335.4 to 413.9)** | **59.0 (52.8 to 65.1)** |
| 0 g | **87.8 (41.1 to 134.5)** | **8.7 (4.1 to 13.3)** | **178.9 (115.6 to 242.3)** | **14.7 (9.52 to 20.0)** | **316.1 (252.7 to 379.4)** | **29.4 (23.5 to 35.3)** |
| Bottled water | 30.3 (−62.0 to 1.4) | 4.2 (−8.7 to 0.2) | **82.1 (−125.7 to −38.4)** | **−10.5 (−16.1 to −4.9)** | **−130.5 (−174.1 to −88.8)** | **−15.7 (−20.9 to −10.4)** |
| Levy exempt drinks | | | | | | |
| Alcoholic drinks | −16.5 (−48.5 to 15.4) | 1.0 (−2.8 to 0.9) | **−84.9 (−135.1 to −34.7)** | **−4.8 (−7.7 to −2.0)** | **−103.1 (−153.3 to −53.0)** | **−5.8 (−8.6 to −3.0)** |
| Milk and milk based drinks* | **−185.5 (−249.7 to −121.4)** | **−4.9 (−6.6 to −3.2)** | **145.5 (64.4 to 226.6)** | **4.2 (1.9 to 6.6)** | **−132.8 (−213.9 to −51.7)** | **−3.6 (−5.7 to −1.4)** |
| No added sugar fruit juices | 6.8 (−6.9 to 20.5) | 1.4 (−1.4 to 4.3) | −6.2 (−24.8 to 12.5) | −1.26 (−6.1 to 2.5) | 8.7 (−9.9 to 27.3) | 1.82 (−2.1 to 5.7) |
| Drinks sold as powders (g) | **−6.9 (−10.0 to −3.8)** | **−6.8 (−9.9 to −3.8)** | **9.6 (5.3 to 13.9)** | **11.2 (6.2 to 16.2)** | 0.9 (−3.3 to 5.2) | 1.0 (−3.5 to 5.5) |
| Other categories | | | | | | |
| Confectionery (g) | −10.1 (−53.9 to 33.8) | −2.4 (−13.1 to 8.2) | 39.8 (−19.0 to 98.6) | 11.6 (−5.5 to 28.8) | 35.3 (94.1 to −23.5) | 10.2 (−6.8 to 27.1) |
| All soft drinks combined (ie, excluding alcohol) | 11.8 (−103.7 to 127.3) | 0.2 (−1.4 to 1.7) | **187.8 (29.7 to 345.9)** | **2.6 (0.4 to 4.7)** | **188.8 (30.7 to 346.9)** | **2.6 (0.4 to 4.7)** |

Bold indicates significant difference at 95% CI level.
*Milk comprises drinks in the following categories: semi-skimmed; specific low fat % milk (eg, 1% fat milk); whole milk; buttermilk; modified milk; other milk; other non-cows milk; rice drink; soya milk. Skimmed milk is excluded from all analysis in this paper due to missing data.

**Table 3** Absolute and relative change in sugar in drinks and confectionery (g) purchased per household (95% CI) per week in relation to the UK soft drinks industry levy, March 2014 to March 2019 (weighted)

| | Analysis 1: preannouncement and postannouncement (March 2014–March 2018) | | Analysis 2: pre-implementation and post implementation (March 2016–March 2019) | | Analysis 3: preannouncement, postannouncement and pre-implementation, post implementation (March 2014–March 2019) | |
|---|---|---|---|---|---|---|
| | Absolute change (g) | Relative change (%) | Absolute change (g) | Relative change (%) | Absolute change (g) | Relative change (%) |
| **All drinks** | | | | | | |
| **Levy liable drinks (sugar/100 mL)** | | | | | | |
| High tier (≥8 g) | **5.5 (3.8 to 7.2)** | **10.8 (7.4 to 14.1)** | **−21.2 (−23.8 to −18.5)** | **−49.3 (−55.4 to −43.1)** | **−16.2 (−18.8 to −13.5)** | **−42.6 (−49.6 to −35.6)** |
| Low tier (≥5 g to <8 g)* | **−4.3 (−6.1 to −2.6)** | **−37.5 (−52.5 to −22.5)** | **−5.0 (−7.4 to −2.6)** | **−75.8 (−112.7 to −38.9)** | **−11.5 (−13.9 to −9.1)** | **−87.8 (−106.4 to −69.2)** |
| No levy (<5 g)† | **5.7 (3.9 to 7.4)** | **72.6 (50.3 to 94.9)** | **9.7 (7.3 to12.1)** | **56.0 (41.9 to 70.0)** | **19.2 (16.7 to 21.6)** | **242.8 (211.9 to 273.7)** |
| >0 g to <5 g | **5.7 (3.9 to 7.4)** | **72.6 (50.3 to 94.9)** | **9.7 (7.3 to12.1)** | **56.0 (41.9 to 70.0)** | **19.2 (16.7 to 21.6)** | **242.8 (211.9 to 273.7)** |
| **Levy exempt drinks** | | | | | | |
| Milk and milk based drinks‡ | **−3.9 (−6.5 to −1.3)** | **−2.2 (−3.6 to −0.7)** | **4.1 (0.5 to 7.7)** | **2.4 (0.3 to 4.6)** | −3.1 (−6.7 to 0.5) | −1.8 (−3.8 to 0.3) |
| No added sugar fruit juices | **2.6 (0.3 to 4.8)** | **5.7 (0.7 to 10.7)** | −1.7 (−4.8 to 1.5) | −3.5 (−10.0 to 3.0) | 2.6 (−0.5 to 5.7) | 5.9 (−1.2 to 13.1) |
| Drinks sold as powders | 0.3 (−1.6 to 2.2) | 1.6 (−7.5 to 10.6) | −0.04 (−2.7 to 2.6) | −0.2 (−13.9 to 13.5) | 1.1 (−1.6 to 3.7) | 5.7 (−8.8 to 20.2) |
| **Other categories** | | | | | | |
| Confectionery | −6.6 (−32.0 to 18.9) | −2.8 (−13.8 to 8.1) | 22.1 (−12.0 to 56.1) | 11.4 (−6.2 to 29.1) | 18.4 (−15.7 to 52.4) | 9.3 (−8.0 to 26.7) |
| All soft drinks combined (ie, excluding alcohol) | **4.6 (0.5 to 8.6)** | **1.4 (0.2 to 2.7)** | **−12.9 (−18.5 to −7.4)** | **−4.3 (−6.1 to −2.4)** | **−8.0 (−13.6 to −2.4)** | **−2.7 (−4.5 to −0.8)** |

Bold indicates significant difference at 95% CI level.
*The counterfactual for low tier drinks crossed 0 mL shortly before the end of the study period thus predicting negative purchases; therefore, the non-counterfactual estimate at the end of the study period was compared with the final week during which the counterfactual was a positive number.
†We do not report change in sugar purchased from drinks with 0 g sugar/100 mL or bottled water as these contain no sugar; the figures for the combined no levy line and the >0 g to <5 g of sugar/100 mL line are the same as the only drinks in the no levy category containing sugar are those with >0 g to <5 g.
‡Milk comprises drinks in the following categories: semi-skimmed; specific low fat % milk (eg, 1% fat milk); whole milk; buttermilk; modified milk; other milk; other non-cows milk; rice drink; soya milk. Skimmed milk is excluded from all analysis in this paper due to missing data.

with the counterfactual of preannouncement trends, in March 2019 volume of alcoholic drinks purchased decreased by 103.1 mL (53.0 to 153.3 mL) per household per week, equivalent to a 5.8% (3.0% to 8.6%) reduction; and volume of milk and milk based drinks purchased decreased by 132.8 mL (51.7 to 213.9 mL), equivalent to a 3.6% (1.4% to 5.7%) reduction.

### Secondary analysis: all soft drinks categories combined

Table 3a in online supplemental material E and figure 2a,b in online supplemental material F summarise the results of the controlled interrupted time series analyses of the associated effects of the SDIL on purchased volume of, and sugar from, all soft drinks categories combined, irrespective of levy eligibility. Table 3b in online supplemental material E summarises absolute and relative changes in volume of, and sugar in, all soft drinks and confectionery purchased. Summary figures are also provided in tables 2 and 3.

Overall, compared with the counterfactual estimated from preannouncement trends, a small increase was observed in volume of all soft drinks purchased in March 2019 of 188.8 mL (30.7 to 346.9 mL) per household per week, equivalent to a 2.6% (0.4% to 4.7%) increase. A reduction was, however, found in sugar purchased in all soft drinks (including exempt drinks) combined of 8.0 g per household per week (2.4 to 13.6 g), equivalent to 2.7% (0.8% to 4.5%).

### Sensitivity analyses

Excluding manufacturers of levy liable products with less than 1 million and less than 500 000 L of purchased drinks annually in our dataset was associated with small changes in the magnitude of estimated coefficients, but with no change in the direction or statistical significance of absolute or relative changes in volume of, or sugar in, drinks (table 4a,b in online supplemental material F).

In general, removing the control category led to minor changes in effect estimates but wider CIs (see table 5a–d in online supplemental material G).

### DISCUSSION

Taking account of pre-existing preannouncement trends, this study found that 1 year after implementation of the SDIL, sugar purchased from all soft drinks combined that were taken home decreased by 8.0 g per household per week (or 2.7%), while volume increased by 188.8 mL per household per week (or 2.6%). Assuming a mean UK household size of 2.4 people,[34] this is equivalent to a reduction in sugar from SSBs of 3.3 g per person per week and an increase in volume of 79 mL per person per week, or equivalent to the replacement of 66 mL of a drink with 5 g sugar per 100 mL per person per week with 145 mL of a sugar-free alternative. A modelling study conducted before implementation of the SDIL found that if the levy achieved reformulation it could be expected to lead to a decrease in sugar consumption from SSBs (from all

sources, not just for consumption at home) of 7–38 g per person per week and that this would be associated with a reduction in the number of obese individuals in the UK of 0.2%–0.9% and a reduction in incident cases of type 2 diabetes of −2.0 to 31.1 per 1000 person years.[35] The reduction in sugar from SSBs we report 1 year after implementation of the SDIL is around half of these lower effect estimates.

### Strengths and weaknesses of this study

In this study we used a large, nationally representative dataset, included a control category, and explored changes in two potential substitute categories (alcohol and confectionery).

We only included purchases brought into homes. Although KWP also collects data on other purchases, this smaller panel was established in mid-2015 and so was unsuitable for our analyses because robust preannouncement trends could not be estimated. KWP data are collected at the household level and do not take account of waste or differential sharing within households. Nevertheless, the data provide a reasonable estimate of consumption.[36] We did not assess changes in other categories beyond soft drinks, alcohol, and confectionery.

The estimate of effect size in interrupted time series analyses is based on a modelled counterfactual that might be inaccurate. For example, the strong downward trend in higher tier drinks before the announcement of SDIL might not have continued. Attribution of effects in interrupted time series analyses is vulnerable to time varying confounding including cointerventions. The SDIL is part of a wider sugar reduction strategy, although this has been found to have achieved minimal changes beyond those attributable to the SDIL.[25]

The personal characteristics of the panel remained similar over the study period, and proprietary weightings were used to account for non-consumers and to adjust for variations in panel composition. Households participating in KWP are slightly more likely to be from lower social grades and to have no qualifications compared with UK households generally. This might reflect the relative value placed on the small rewards for participation by different households and could limit the generalisability of our findings. If households from lower socioeconomic backgrounds are more likely to change purchasing as a result of the SDIL, then we could have marginally overestimated the effect of the SDIL. However, while we previously found that the price of soft drinks in the UK did change after implementation of the SDIL, no clear pattern was found, with the price of some groups of drinks increasing and others decreasing.[15] We previously found no systematic differences between the sugar content of drinks reported in KWP data and contemporaneous values listed on supermarket websites.[27]

### Comparison with other work

Our finding that the SDIL was associated with a reduction in purchased sugar from all soft drinks combined

is consistent with previous analyses that focused on the SDIL.[24 25] Although our estimate of the reduction in sugar consumption from all soft drinks combined associated with the levy (2.7%) is less than that estimated by others (29%)[25] this previous estimate did not take account of pre-existing trends which we have demonstrated were on a steep downward trajectory for high tier drinks.

We found that the reduction in purchased sugar from all soft drinks combined alongside a 2.6% increase in volume of all soft drinks purchased. This is consistent with previously reported reductions in the sugar concentration of drinks associated with the SDIL.[15] However, the estimated effect size is below the range of reformulation scenarios modelled before implementation (ie, a reduction of 17–90 g of sugar per household per week).[35] This difference may be, at least partly, attributable to our focus on drinks taken home versus the modelling study's focus on all drinks. Furthermore, the modelling was based on pre-implementation best and worst case scenarios of changes in formulation, price and SSB market share while our analysis was based on observed data.

Evaluations of other SSB taxes have revealed a consistent trend of reductions in purchasing of taxed drinks and no change in purchasing of untaxed drinks.[5] We found similar with both volume of, and sugar in, high and low tier drinks decreasing overall. However, these reductions in volume of taxed drinks were more than offset by increases in volume of no levy drinks purchased. Despite some increases in sugar purchased in no levy drinks, these did not offset decreases in sugar purchased from high and low tier drinks. The SDIL is relatively unique in being explicitly designed to encourage reformulation and there is evidence that substantial reformulation occurred.[15] We are not able to determine from our findings whether the changes we report are due to changes in consumer preference, formulation, or both.

### Meaning of the study and implications for policymakers

Our main findings are that the SDIL was associated with a reduction in purchased sugar from all soft drinks combined with evidence of an increase in the total volume of all soft drinks purchased. Given the reformulation associated with the SDIL already documented,[15] it is probable that the changes we report were driven by reductions in the sugar concentration of available drinks, alongside consumers switching to and indeed increasing consumption of, lower sugar alternatives. Despite the overall reduction we found in sugar purchased in soft drinks, the average amount of sugar purchased in drinks in the no levy group paradoxically increased after implementation of the SDIL, with many drinks that previously had sugar concentrations above the levy threshold now having them just below the threshold. This seems to reflect manufacturers reformulating to target thresholds. Lowering the threshold sugar concentration at which drinks become eligible for the SDIL even further could potentially lead to greater overall reductions in sugar concentrations and sugar purchased in soft drinks, as could extension of the

SDIL to milk based drinks and other currently exempt categories that sometimes contain high levels of sugar.

The SDIL has also been found to have had no long-term negative effects on the share value or turnover of UK soft drinks companies,[37 38] suggesting that contrary to industry predictions, public health can gain without negatively affecting the soft drinks sector.

We note a marked pre-implementation decline in purchasing of high levy tier drinks. It is possible that this was, at least in part, driven by concern from industry about a possible SSB tax, leading to some preannouncement reformulation; alongside growing consumer awareness of, and concerns about, the health impacts of SSBs.[39] Although it is uncertain if this trend would have continued in the absence of the SDIL, it is likely to be beneficial for health.

Reassuringly, we did not observe any increase in purchasing of potentially harmful substitutes (ie, alcohol and confectionery) associated with the SDIL, which could have partially or wholly offset any public health gains from the SDIL. However, we did not study the SDIL's effect on purchases of other food groups or on overall diet.

In contrast with previous findings from Mexico and Barbados,[6 40] we did not observe an increase in purchased bottled water associated with the SDIL. Indeed purchases of bottled water decreased significantly during the study period (by 130.5 mL per household per week, or 15.7%). Although we cannot rule out an effect of the SDIL on bottled water purchases, we cannot think of a plausible pathway through which it achieved reductions in purchased bottled water. Instead, this reduction might be due to coincident increases in concern about single use plastic that have been attributed, in the UK, to the broadcast of the nature documentary series *Blue Planet 2* in October to December 2017.[41] It is not clear if a similar 'Blue Planet effect' has occurred in other countries. Unlike for many other soft drinks, a like-for-like substitution is available for bottled water in countries such as the UK—that is, filling reusable water bottles with tap water. Several UK retailers have reported substantial growth in sales of reusable water bottles since 2018.[42] Given that tap water is freely available, it is difficult to study changes in its consumption directly.

### Unanswered questions and future research

Future work should seek to understand the longer term effects of the SDIL on purchasing and consumption of soft drinks as well as total diet, and health outcomes. Differential effects of the SDIL on all these outcomes across population groups (eg, by socioeconomic position and in households with vs without children) should also be explored to determine whether the SDIL contributes to narrowing inequalities in health. The changes in purchasing we report here could be used as an input to health impact modelling to estimate the effect of changes on population prevalence of obesity, diabetes and other chronic conditions to determine how apparently small changes in consumption at the household level translate

into health benefits. It is likely that the reformulation that has occurred in response to the SDIL[15] reflects substantial increases in the use of artificial sweeteners in the UK soft drinks market. Given public mistrust of artificial sweeteners[39] and the recent advice from WHO that artificial sweeteners should not be used to reduce the risk of non-communicable diseases,[43] the effect of the SDIL on consumption of these should also be explored.

## CONCLUSION

One year after implementation of the SDIL, purchased sugar in all soft drinks combined decreased by around 8 g per household per week (or 2.7%) with an increase in the volume of purchased soft drinks of 189 mL per household per week (or 2.6%). Further studies are required to determine whether and how these apparently small effect sizes translate into health outcomes.

**Author affiliations**
[1]MRC Epidemiology Unit, Cambridge, UK
[2]University College London Great Ormond Street Institute of Child Health, London, UK
[3]York University - Keele Campus, Toronto, Ontario, Canada
[4]Nuffield Department of Population Health, University of Oxford, Oxford, UK
[5]Institute for Health Policy and Clinical Practice, University of Warwick Warwick Medical School, Coventry, UK
[6]Department of Social and Environmental Health Research, London School of Hygiene & Tropical Medicine, London, UK
[7]NIHR Oxford Biomedical Research Centre, Oxford, UK
[8]University of Bath Department of Social and Policy Sciences, Bath, UK
[9]Nuffield Department of Population Health, Centre on Population Approaches for Non-Communicable Disease Prevention, Oxford, UK
[10]University of Exeter, Exeter, UK

**Contributors** DP, MW, SC, MR, RS, HR, JA, PS, OTM, AB and TLP conceived the study and defined the analytical strategy. DP, NR, JA, OTM, TLP and SS performed statistical analyses. DP, NR, JA, OTM, TLP, CP-J and SS provided preliminary interpretation of findings. DP, NR, JA, OTM and TLP drafted the manuscript. All authors critically interpreted the results, revised the manuscript, provided relevant intellectual input, and read and approved the final manuscript. NR and JA had primary responsibility for the final content. JA will act as guarantor. The corresponding author attests that all listed authors meet authorship criteria and that no others meeting the criteria have been omitted.

**Funding** NR, OTM, MW and JA were funded by the Centre for Diet and Activity Research (CEDAR), a UK Clinical Research Collaboration (UKCRC) Public Health Research Centre of Excellence Funding from the British Heart Foundation, Cancer Research UK, Economic and Social Research Council, Medical Research Council, National Institute for Health Research (NIHR) and Wellcome Trust, under the auspices of the UKCRC. This project was funded by the NIHR Public Health Research programme (grant numbers 16/49/01 and 16/130/01), and these grants supported purchase of the data used in this study. The work was also supported by the Medical Research Council (grant numbers MC_UU_12015/6 and MC_UU_00006/7). The views expressed are those of the authors and not necessarily those of the National Health Service, the NIHR, or the Department of Health and Social Care, UK. The funders had no role in study design, data collection and analysis, decision to publish, or preparation of the manuscript. NR and JA had full access to all the data in the study. All authors had final responsibility for the decision to submit for publication. The lead author affirms that the manuscript is an honest, accurate, and transparent account of the study being reported; that no important aspects of the study have been omitted; and that any discrepancies from the study as planned (and, if relevant, registered) have been explained.

**Competing interests** MW was director of the National Institute for Health Research Public Health Research Funding programme when this work was conducted and OTM was on secondment at the UK Department of Health and Social Care when this work was conducted and previously worked with Public Health England.

**Patient and public involvement** Patients and/or the public were involved in dissemination plans of this research. Refer to the Methods section for further details.

**Patient consent for publication** Not applicable.

**Provenance and peer review** Not commissioned; externally peer reviewed.

**Data availability statement** Data may be obtained from a third party and are not publicly available. The statistical code for the analyses is available from https://github.com/MRC-Epid/SDILEvaluation. Kantar Worldpanel data are not publicly available but can be purchased from Kantar Worldpanel (http://www.kantarworldpanel.com). The authors are not legally permitted to share the data used for this study but interested parties can contact Kantar WorldPanel representative Sean Cannon (Sean.Cannon@kantar.com) to inquire about accessing this proprietary data. The lead author affirms that the manuscript is an honest, accurate and transparent account of the study being reported; that no important aspects of the study have been omitted; and that any discrepancies from the study as planned (and, if relevant, registered) have been explained.

**ORCID iDs**
David Pell http://orcid.org/0000-0002-9357-6804
Oliver T Mytton http://orcid.org/0000-0003-3218-9912
Steven Cummins http://orcid.org/0000-0002-3957-4357
Harry Rutter http://orcid.org/0000-0002-9322-0656
Peter Scarborough http://orcid.org/0000-0002-2378-2944
Jean Adams http://orcid.org/0000-0002-5733-7830

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
