## [Reviewer comments · BMJ Open]

PEER REVIEW HISTORY

This article is a corrected version of Pell 2021 published in The BMJ, which has now been retracted. The previous versions of the manuscript along with the peer review of Pell 2021, which informed the publication of the BMJ Open article, are available here <https://www.bmj.com/content/372/bmj.n254/peer-review>.

BMJ Open publishes all reviews undertaken for accepted manuscripts. Reviewers are asked to complete a checklist review form (<http://bmjopen.bmj.com/site/about/resources/checklist.pdf>) and are provided with free text boxes to elaborate on their assessment. These free text comments are reproduced below.

ARTICLE DETAILS

TITLE (PROVISIONAL)	Changes in soft drinks purchased by British households associated with the UK soft drinks industry levy: a controlled interrupted time series analysis
AUTHORS	Rogers, Nina; Pell, David; Mytton, Oliver; Penney, Tarra; Briggs, Adam; Cummins, Steven; Penn-Jones, Catrin; Rayner, Mike; Rutter, Harry; Scarborough, Peter; Sharp, Stephen; Smith, Richard; White, Martin; Adams, J

VERSION 1 – REVIEW

REVIEWER	Vigilla-Montecillo, Kristine
REVIEW RETURNED	28-Jul-2023

GENERAL COMMENTS	The research question is well-defined, making it easy to understand the purpose of the study. The analysis is suitable for addressing the research question and evaluating the changes over time. The study assesses not only the volume of drinks purchased but also the amount of sugar in different drink categories, providing a comprehensive view of the impact of the SDIL. The results are presented clearly, with specific numbers and confidence intervals, making it easy to interpret the findings.
---

REVIEWER	Warendorf, Eve University of Applied Sciences Esslingen Faculty of Social Work Health and Nursing
REVIEW RETURNED	03-Aug-2023

GENERAL COMMENTS	Patient and public reviewers do not need medical or scientific training. What we seek and value from you is your perspective on a slightly different set of questions than those posed to traditional peer reviewers. They include the following: • Are the questions the paper addresses relevant and important to patients and/or carers? The questions the paper addresses are relevant and important to patients/carers. Obesity is rising all over the world with exposure to the Western diet and the government needs to counter purchasing of tasty unhealthy food. So, it is good to discover the effects of a levy. • Are there topics or issues that are missing, or need to be highlighted more?
--

Sugar decreased by 8 g teaspoons a week and I wonder if the effort of the levy is worth the
per household per result. If the authors have the data, I would like to know if some
week. This is only 2

	groups of patients benefited more than others. For example, did obese patients decrease their sugar consumption by more than patients of normal weight? I really would like more discussion, because 2.7% is now 10 times less than that estimated by others and it seems a very small benefit for all the work! Why would other studies show 29%? What about the implementation could have caused the difference? I have been surprised by drinks from bottled water manufacturers which have the sweetness of soft drinks. I wish that they would reformulate for less sweetness, but I suppose reformulation is generally not changing the sweetness by adding natural and artificial sweeteners which are not sugar. Is the reduction in sugar worth the addition of sweeteners which may not be entirely healthy?  • Is the treatment or intervention suggested or guidance given something which patients/carers can readily take up? or does it present challenges? The levy means that some patients will change their soft drink habits to save money or after tasting all the new, improved soft drinks. Awareness may make them label-readers and cause them to make healthier choices.  • Are the outcomes described/measured in the study important to patients/carers? Are there others that should have been considered? The outcomes are important to patients/carers, because reduction of sugar in soft drinks could be a way for patients to maintain or lose weight without effort.  • Do you have any suggestions that might help the author(s) strengthen their paper and make it more useful for doctors to share and discuss with patients/ carers? • Do you think the level of patient/carer involvement in the study could have been improved? If there was none do you have ideas on how they might have done so? The level of patient/carer involvement could have been improved. The authors could have asked people in different age groups and with different BMI about their habits and the importance of price, taste, perceived healthiness. From <https://www.bmj.com/about-bmj/resources-reviewers/guidance-patient-reviewers>
--	--

REVIEWER	Jones, Alexandra The George Institute for Global Health
REVIEW RETURNED	13-Aug-2023

GENERAL COMMENTS	GENERAL COMMENTS This paper provides an analysis of the volume of, and sugar content of, sugary drinks sold before and after introduction of the United Kingdom's Soft Drinks Industry Levy (SDIL). This is a revised version of a paper I reviewed for the BMJ in 2020. In comparison to the original paper, this revision reports a smaller
---

	reduction in sugar purchased from all drinks, and a small increase in the volume of soft drinks bought before and after the SDIL. The paper remains important given continued interest in SSB taxes worldwide, and worldwide interest in the UK SDIL design which is often cited as 'best practice'. While disappointing that the results are less significant, and as discussed by the authors are less than were perhaps predicted, the results remain positive in that they demonstrate sugar reductions that could benefit health and also sales results that aren't harmful (and may in this new version actually benefit) industry. It is important that this paper is published in its corrected version. The study has been designed and carried out by a team of recognised experts who have published numerous other pieces of evidence in this area, and the paper is very well-written. I have only minor comments on the revised manuscript text to improve clarity and round out discussion around the policy implications of the results. I have read the track changed manuscript and note some small areas of change (e.g. mention of Kantar weightings, slightly smaller number of families included), and remain curious about the cause of the change in results though acknowledge this may not be content that the authors wish to include in this revised version. I (Dr Alexandra Jones) make the following comments based on my experience in food policy. My research applies a mix of quantitative, legal and policy analysis, with outcomes used to strategically inform policy and improve regulation of unhealthy products. That may make me a reasonable end-user of this research. However, as noted in my previous review, my primary expertise is not in statistical methods, thus I am relying upon other reviewers to provide a more detailed critique of this aspect of the paper. ABSTRACT, STRENGTHS AND LIMITATIONS BULLETS, INTRO All are clearly written. In the strengths and limitations line 49 final bullet punctuation towards the end of this sentence could improve clarity – I was unsure how to read it. Very minor/potentially stylistic but I would prefer the intro to finish with more explicit use of the word aim/objective rather than 'we determined whether...' and words to make it clear that you have ended the summary of the existing work and started talking about what this paper will add to that. METHODS As per my comments on the original version, the study design is appropriate and adequately designed to answer the research question. The comparison between levied drink categories and unlevied products (which would be a desirable substitution), plus exempt categories like alcohol, and confectionery (which would be an undesirable substitution) add to the validity of the findings. The main outcome measure – volume purchased per household per week of each drink category of interest, and the sugar
--	--

	purchased per household per week from each of those categories – was clear. RESULTS The results are clearly presented in both the text and figures. The findings are credible. The results in this updated analysis have changed from the previous version, showing a reduction in sugar of 8g per household per week, or 2.7% instead of 30g a week, or 10%. There is also a small increase in the volume (compared to no change in sales volume in previous version) of all soft drinks sold. DISCUSSION The discussion is appropriately derived from the data. It is notable that the authors now acknowledge that their findings are less than half of estimated effect sizes from modelling done before the SDIL was implemented. They do note that the fact that Kantar data relates only to purchases brought home (as opposed to drinks consumed out in entertainment settings for example) could be one reason for this, though it might be useful to also suggest any other explanations they think are likely given the change in results from the earlier work. Near the close of the discussion (Page 16, line 7) the authors talk about likely reformulation behind the change in sugar content and note ‘given consumers mistrust of sweeteners’ the effect of SDIL on consumption of these should be explored. I think it should be noted that it is not just consumer mistrust – in light of recent work at WHO I think it would be good to see a reference to the new WHO Guideline on non-sugar sweeteners and say something about how the design of the SDIL might need to be revisited in time if ongoing consumption of artificial sweeteners also isn’t good for long-term health. CONCLUSION The conclusion is appropriate- I note that it is essentially the same as before as the findings do point in the same direction, just not as strongly (at least for the health results, though the benefit to industry might actually be better than previously thought given sales results).
--	---

REVIEWER	Wade, Angie UCL
REVIEW RETURNED	12-Sep-2023

GENERAL COMMENTS	This paper has been extensively reviewed and discussed when submitted to the BMJ and at the decision to retract. The authors (updated) have made changes in line with those discussed at a meeting with concerned parties (senior statistician on the paper not present). I am happy that the authors have implemented the changes adequately, including updated graphics as mooted.
--

VERSION 1 – AUTHOR RESPONSE

In response to Reviewer: 1

The research question is well-defined, making it easy to understand the purpose of the study. The analysis is suitable for addressing the research question and evaluating the changes over time. The study assesses not only the volume of drinks purchased but also the amount of sugar in different drink categories, providing a comprehensive view of the impact of the SDIL. The results are presented clearly, with specific numbers and confidence intervals, making it easy to interpret the findings.

Thank you for this positive review.

In response to Reviewer: 2

- Are the questions the paper addresses relevant and important to patients and/or carers?

The questions the paper addresses are relevant and important to patients/carers. Obesity is rising all over the world with exposure to the Western diet and the government needs to counter purchasing of tasty unhealthy food. So, it is good to discover the effects of a levy.

Thank you for these positive comments

- Are there topics or issues that are missing, or need to be highlighted more?

Sugar decreased by 8 g per household per week. This is only 2 teaspoons a week and I wonder if the effort of the levy is worth the result. If the authors have the data, I would like to know if some groups of patients benefited more than others. For example, did obese patients decrease their sugar consumption by more than patients of normal weight?

Unfortunately information on the weight status of participants included in the data set used in this paper is not collected. However, we have conducted other analyses using data from the National Child Measurement Programme that indicate that implementation of the Soft Drinks Industry Levy (SDIL) was associated with a decrease in obesity prevalence particularly in girls in year 6 (aged 10-11 years).[1] Other work exploring socio-economic inequalities is ready for submission.

I really would like more discussion, because 2.7% is now 10 times less than that estimated by others and it seems a very small benefit for all the work! Why would other studies show 29%? What about the implementation could have caused the difference?

As described in our paper, we think one important explanation of this discrepancy is that the study that estimated an effect of 29% did not take into account background trends in purchasing. As there was a steep decrease in purchasing trends over time even before the levy was introduced, a simple before-after study is likely to estimate a much larger effect of the levy than our analysis, which estimates the additional effect of the levy over and above the effect of background trends. Nevertheless, we believe that a 2.7% reduction is not necessarily a 'small' benefit as our modelling suggests that this would translate into a £12.2bn cost benefit to the NHS.

I have been surprised by drinks from bottled water manufacturers which have the sweetness of soft drinks. I wish that they would reformulate for less sweetness, but I suppose reformulation is generally not changing the sweetness by adding natural and artificial sweeteners which are not sugar. Is the reduction in sugar worth the addition of sweeteners which may not be entirely healthy?

Reviewer 3 also touches on this point, particularly with respect to recent guidance from WHO on use of artificial sweeteners. As described below, we have noted the recent WHO guidance about the value of artificial sweeteners in non-communicable disease prevention: *"Given public mistrust of artificial sweeteners, and the recent advice from WHO that artificial sweeteners should not be used to reduce the risk of non-communicable diseases, the effect of the SDIL on consumption of these should also be explored."*

- Is the treatment or intervention suggested or guidance given something which patients/carers can readily take up? or does it present challenges?

The levy means that some patients will change their soft drink habits to save money or after tasting all the new, improved soft drinks. Awareness may make them label-readers and cause them to make healthier choices.

Thank you for these comments.

- Are the outcomes described/measured in the study important to patients/carers? Are there others that should have been considered?

The outcomes are important to patients/carers, because reduction of sugar in soft drinks could be a way for patients to maintain or lose weight without effort.

Thank you for this assessment – we agree.

- Do you have any suggestions that might help the author(s) strengthen their paper and make it more useful for doctors to share and discuss with patients/ carers?
- Do you think the level of patient/carer involvement in the study could have been improved? If there was none do you have ideas on how they might have done so?

The level of patient/carer involvement could have been improved. The authors could have asked people in different age groups and with different BMI about their habits and the importance of price, taste, perceived healthiness.

Thank you for this suggestion. Whilst we have not explored the specific issue of how age, BMI and habits are associated with the importance of price, taste and perceived healthiness, we have conducted other work exploring support for the levy.[2, 3] This found high support for the levy that did not change over the first 20 months of implementation. We also found that support for the levy was greater in people who were older, did not consume SSBs, did not have social norms to drink SSBs, knew about the link between SSBs and obesity and trusted messages from health experts.

In response to Reviewer: 3

This paper provides an analysis of the volume of, and sugar content of, sugary drinks sold before and after introduction of the United Kingdom's Soft Drinks Industry Levy (SDIL).

This is a revised version of a paper I reviewed for the BMJ in 2020. In comparison to the original paper, this revision reports a smaller reduction in sugar purchased from all drinks, and a small increase in the volume of soft drinks bought before and after the SDIL.

The paper remains important given continued interest in SSB taxes worldwide, and worldwide interest in the UK SDIL design which is often cited as 'best practice'. While disappointing that the results are less significant, and as discussed by the authors are less than were perhaps predicted, the results remain positive in that they demonstrate sugar reductions that could benefit health and also sales results that aren't harmful (and may in this new version actually benefit) industry. It is important that this paper is published in its corrected version.

The study has been designed and carried out by a team of recognised experts who have published numerous other pieces of evidence in this area, and the paper is very well-written.

Thank you for this summary. We agree that the results demonstrate that the levy could benefit health and that it is unlikely to be harmful to industry.

I have only minor comments on the revised manuscript text to improve clarity and round out discussion around the policy implications of the results. I have read the track changed manuscript and note some small areas of change (e.g. mention of Kantar weightings, slightly smaller number of families included), and remain curious about the cause of the change in results though acknowledge this may not be content that the authors wish to include in this revised version.

This suggestion was also made by the editor. As described in response to the editorial comments above, we have added a statement at the end of the methods section describing the previous error made and that this has been corrected in the current version: *"This paper is a corrected version of, now retracted, Pell et al (2021), originally published in the BMJ. The analysis presented in the original Pell et al (2021) paper included an incorrect weighting variable. This variable was incorrectly calculated as the inverse of what it should have been. The variable was also redundant to the analysis as it replicated a component of a second weighting variable also included (the "proprietary weights provided by KWP" mentioned above). The current corrected version replicates the original analysis without this redundant and incorrectly calculated weighting variable. The second, correct, weighting variable (the*

“proprietary weights provided by KWP” mentioned above) remains included. The authors identified the error themselves and alerted the journal and readers.”

ABSTRACT, STRENGTHS AND LIMITATIONS BULLETS, INTRO

All are clearly written.

Thank you.

In the strengths and limitations line 49 final bullet punctuation towards the end of this sentence could improve clarity – I was unsure how to read it.

We have clarified this bullet point so that it now reads: *“Attribution of effects in interrupted time series analyses is vulnerable to time varying confounding such as co-interventions.”*

Very minor/potentially stylistic but I would prefer the intro to finish with more explicit use of the word aim/objective rather than ‘we determined whether...’ and words to make it clear that you have ended the summary of the existing work and started talking about what this paper will add to that.

We have revised the final sentence of the introduction to read: *“In this paper our aim was to determine whether household purchases of drinks and confectionery had changed one year after implementation of the SDIL.”*

METHODS

As per my comments on the original version, the study design is appropriate and adequately designed to answer the research question. The comparison between levied drink categories and unlevied products (which would be a desirable substitution), plus exempt categories like alcohol, and confectionery (which would be an undesirable substitution) add to the validity of the findings.

The main outcome measure – volume purchased per household per week of each drink category of interest, and the sugar purchased per household per week from each of those categories – was clear.

Thank you for these positive comments.

RESULTS

The results are clearly presented in both the text and figures. The findings are credible.

The results in this updated analysis have changed from the previous version, showing a reduction in sugar of 8g per household per week, or 2.7% instead of 30g a week, or 10%.

There is also a small increase in the volume (compared to no change in sales volume in previous version) of all soft drinks sold.

Thank you for this accurate summary.

DISCUSSION

The discussion is appropriately derived from the data. It is notable that the authors now acknowledge that their findings are less than half of estimated effect sizes from modelling done before the SDIL was implemented. They do note that the fact that Kantar data relates only to purchases brought home (as opposed to drinks consumed out in entertainment settings for example) could be one reason for this, though it might be useful to also suggest any other explanations they think are likely given the change in results from the earlier work.

We have clarified that *“the modelling was based on pre-implementation best and worst case scenarios of changes in formulation, price and SSB market share whilst our analysis was based on observed data.”* One other reason for discrepancies is inaccuracies of the predictions made to generate the scenarios in the scenario modelling.

Near the close of the discussion (Page 16, line 7) the authors talk about likely reformulation behind the change in sugar content and note ‘given consumers mistrust of sweeteners’ the effect of SDIL on consumption of these should be explored. I think it should be noted that it is not just consumer mistrust – in light of recent work at WHO I think it would be good to see a reference to the new WHO Guideline on non-sugar sweeteners and say something about how the design of the SDIL might need to be revisited in time if ongoing consumption of artificial sweeteners also isn’t good for long-term health.

We have extended this sentence to capture this point: *“Given public mistrust of artificial sweeteners, and the recent advice from WHO that artificial sweeteners should not be used to reduce the risk of non-communicable diseases, the effect of the SDIL on consumption of these should also be explored.”*

CONCLUSION

The conclusion is appropriate - I note that it is essentially the same as before as the findings do point in the same direction, just not as strongly (at least for the health results, though the benefit to industry might actually be better than previously thought given sales results).

Thank you. We agree that the conclusion is appropriate as the direction of effect for health, in particular, is unchanged.

In response to Reviewer: 4

This paper has been extensively reviewed and discussed when submitted to the BMJ and at the decision to retract. The authors (updated) have made changes in line with those discussed at a meeting with concerned parties (senior statistician on the paper not present). I am happy that the authors have implemented the changes adequately, including updated graphics as mooted.

Thank you for these supportive comments.

References cited in this response

1. Rogers NT, Cummins S, Forde H, Jones CP, Mytton O, Rutter H, et al. Associations between trajectories of obesity prevalence in English primary school children and the UK soft drinks industry levy: An interrupted time series analysis of surveillance data. *PLoS Med.* 2023;20(1):e1004160. doi: 10.1371/journal.pmed.1004160.
2. Pell D, Penney T, Hammond D, Vanderlee L, White M, Adams J. Support for, and perceived effectiveness of, the UK soft drinks industry levy among UK adults: cross-sectional analysis of the International Food Policy Study. *BMJ Open.* 2019;9(3):e026698. doi: 10.1136/bmjopen-2018-026698.
3. Adams J, Pell D, Penney T, Hammond D, Vanderlee L, White M. Public acceptability of the UK Soft Drinks Industry Levy: repeat cross-sectional analysis of the International Food Policy Study (2017–2019). *BMJ Open.* 2021;11(9):e051677. doi: 10.1136/bmjopen-2021-051677.